# Using the TUG Test for the Functional Assessment of Patients with Selected Disorders

**DOI:** 10.3390/ijerph19084602

**Published:** 2022-04-11

**Authors:** Krzysztof Graff, Ewa Szczerbik, Małgorzata Kalinowska, Katarzyna Kaczmarczyk, Agnieszka Stępień, Małgorzata Syczewska

**Affiliations:** 1Department of Pediatric Rehabilitation, Children’s Memorial Health Institute, 04-730 Warsaw, Poland; k.graff@ipczd.pl (K.G.); e.szczerbik@ipczd.pl (E.S.); m.kalinowska@ipczd.pl (M.K.); m.syczewska@ipczd.pl (M.S.); 2Faculty of Rehabilitation, Józef Piłsudski University of Physical Education in Warsaw, 00-968 Warsaw, Poland; agnieszka.stepien@awf.edu.pl

**Keywords:** TUG test, percentage of individual components of the TUG test, children

## Abstract

One of the tests used for quantitative diagnostics is Timed Up-and-Go (TUG), however, no reports were found regarding the percentage share of individual test components, which seems to have a greater diagnostic value in differentiating the functional status of the patients. The aim of the study was to analyze the percentage of the individual components of the TUG test in functional assessment in a population of healthy children and in clinical trials patients with various diseases. Material and Methodology. The material consisted of patients with orthopedic (*n* = 165), metabolic (*n* = 116) and neurological dysfunctions (*n* = 96). Results. The components of the TUG test that differentiated the studied groups of patients to the greatest extent were in the order: relapse tug3%, initial transition tug2%, sitting tug5% and standing up tug1%, while during the final transition tug4% statistically significant differences were found only between healthy children and the studied groups of patients. Conclusions. The TUG test turned out to be a good diagnostic tool, differentiating the studied groups of patients. The analysis of the percentage of the components of the TUG test can help in assessing the mobility of children and adolescents, monitor the effects of physiotherapy or the effects of surgical procedures.

## 1. Introduction

One of the basic diagnostic activities of physiotherapists is to monitor the functional status during the rehabilitation process. Today physiotherapeutic diagnostics increasingly resort to measurement methods to assess the functional state of the musculoskeletal system. These methods are applied to objectively assess the baseline status of patients starting treatment and to monitor the rehabilitation process. The following tests are used for quantitative diagnostics: Timed Up-and-Go (TUG) [1], Eurofit physical fitness test battery [2,3], fitness assessment using a 6-min corridor test [4], and Fullerton test [5].

The second group of measurements are the subjective qualitative scales, e.g., the pain scale according to Wong-Baker, the Tinetti balance test, the Tinetti gait test [6], the Berg balance scale and questionnaires for testing physical activity [7]. The TUG test has been used by numerous authors, among others, for the functional assessment of patients with acquired brain injury [8], for biomechanical analysis during the TUG test of patients with Down’s syndrome [9], for the assessment of the home exercise program for physical fitness and function after completion of chemotherapy in children and adolescents diagnosed with acute lymphoblastic leukemia (ALL) [10], or for the creation of reference database for school-age children [1,11]. Nevertheless, whenever the subject literature quotes the application of the TUG test, it only refers to its total time. No reports were found regarding the percentage share of individual test components, which seems to have a greater diagnostic value in differentiating the functional status of the examined patients.

The objective of the present study was to analyze the percentage of the individual components of the TUG test in functional diagnostics in the population of healthy children and in clinical trials of patients with various diseases.

## 2. Materials and Methods

### 2.1. Research Material

In total 621 people participated in the research. The group of patients consisted of 377 patients of the Rehabilitation Clinic of the Children’s Memorial Health Institute (IP-CZD) in Warsaw, including: 165 children with scoliosis (Sc), 27 patients after surgical treatment of a brain tumor (BT), 14 post-stroke patients (CVA), 15 patients with traumatic brain injury (TBI), 16 children with Guillain–Barré syndrome (GBS), 24 patients with cerebral palsy (CP), 82 patients with morbid obesity (OB) and 34 patients with type I osteogenesis imperfecta (OI). All patients walked independently and were assessed according to the routine procedures adopted at the IP-CZD, to which the parents of the patients gave their consent with a signature placed in the child’s medical history. The control group (*n* = 244) consisted of healthy children (N). Research on patients was approved by the Bioethics Committee of IP-CZD STRATEGMED3/306011/1/NCBR/2017, and the control group studies were approved by the Senate Committee on Ethics of Scientific Research of the AWF SKE 01-02/2019. The detailed clinical characteristics of the research group are presented in Table 1.

### 2.2. Test Procedure

Bodyweight and height were measured in all patients and their calendar age was calculated. Additionally, in patients, based on their medical diagnosis, the side of paresis was determined and a physiotherapeutic examination assessed spine and thoracic deformities, disturbances in the axis of the upper and lower limbs, foot deformities and periarticular contractures. Then, the TUG test was performed with all subjects, considering methodological aspects [12]. The height of the chair was individually adjusted to the height of the tested child. The time of the test was measured from the moment the back of the patient lifted off the back of the chair, till the back was again rested against the back of the chair. Before the test was performed, each subject was informed about how to perform the test, then a demonstration of the test was delivered by the therapist, and then the test was attempted by the patient without time measurement. Patients performed the test five times with the recommendation to “walk as fast as they can” without running. The result of the individual test components was averaged, and the sum of the components represented the total test time. The individual components of the test were: dynamic rising from the chair (tug1), passing the initial 3 m section (tug2), turn (tug3), covering the 3 m section again (tug4) and sitting down with one’s back against the chair’s backrest (tug5). All subjects performed the TUG test by walking around a cone at the end of the 3-m section with the right side of their body. The research was carried out by the same therapist measuring the time of each component of the test (tug1–tug5) on a Samsung Galaxy A70 mobile phone with an accuracy of hundredths of a second. The use of smartphone applications in time measurements in clinical settings is gaining popularity, as they demonstrate high accuracy and consistence with the traditional, lab-based methods [13,14]. In all respondents, the percentage of the individual components of the TUG test was calculated according to the formula:tug_n_% = (tug_n_/TUG)*100(1)
tug_n_%—percentage of the test component, and tug_n_—individual component of the TUG test.

### 2.3. Statistical Analysis

The collected data from the TUG tests were recorded in a spreadsheet and statistical calculations were performed using the STATISTICA 13.3 software suite. The Shapiro–Wilk test was used to assess the normality of the distribution of individual components of the TUG test. Differences in the TUG test results between girls and boys in the control group were assessed using the Kolmogorov–Smirnov test. To assess the link between the TUG test and age, we performed Spearman’s rank-order correlation. The comparison between the investigated groups of diagnoses was performed using the Kruskal–Wallis ANOVA rank test. For all these statistical analyses, we adopted the level of significance alpha = 0.05. The reliability of the TUG test was assessed using the ICC repeatability index (interclass correlation coefficient) for the examiner with the use of the MedCalc program—the ANOVA Kruskal–Wallis test.

## 3. Results

Measurement of the total TUG time by the same therapist was rated at ICC = 0.96. There were significant differences between the groups of patients at the level of (*p* = 0.001) in terms of their body weight, height, and age. Based on Spearman’s rank correlation, no significant correlation was found between the age of the subjects and the TUG test (R = −0.12), therefore further analysis did not consider the division of the studied population into age subgroups. The normal distribution with the Shapiro–Wilk test was found different from the normal one in most cases, therefore non-parametric tests were applied for the analysis. The percentage share of the individual components of the TUG test revealed no significant differences between girls and boys from the control group, therefore the results were considered jointly for both genders. The analysis of the total TUG time demonstrated statistically significant differences between healthy children and all patient groups at the level of *p* = 0.001. There were no significant differences within the studied groups of patients in the total duration of the TUG test. The percentage share of the individual components of the TUG test was adopted for further analysis. Based on the Kruskal–Wallis ANOVA test, significant differences were revealed between the percentages of individual components of the TUG test in the group of patients with neurological diseases. Therefore, this group was divided into two subgroups: NP1 (patients diagnosed with: cerebral palsy (CP) and Guillain–Barré syndrome (GBS), and NP2 (patients with damage to the central nervous system (CNS), including: brain tumor surgery (BT), post-stroke (CVA) and traumatic brain injury (TBI) patients). The obtained TUG test times and their individual components in the target groups of patients are presented in Table 2.

Descriptive data of the percentage of the individual components of the TUG test were characterized considering median, minimum, maximum and 1st and 3rd quartiles as well as the 10th and 90th percentiles, the results are presented in Table 3.

The component that differentiated the groups to the greatest extent was tug3% (turn) and the tug2% component (initial section), while the tug4% component (final section) demonstrated significant differences only between healthy children and patients with scoliosis. Detailed results of the statistical analysis are presented in Table 4.

## 4. Discussion

The repeatability of the TUG test in own research (ICC = 0.96) [15] was similar to the results by other authors [16,17]. So far, the analysis of the percentage of individual components of the TUG test was not applied to assess the gait and balance of pediatric patients with various diseases. The study of Caronni et al. [18] revealed, that in adult neurological patients the turning phase of the TUG test highly correlated with the balance problems and was a good predictor of the Mini-BESTest scale result. In the obese adult women, the walk and turning phases as well as the total TUG time are increased in respect to the women with normal weight [19]. The functional assessment of patients after hip arthroplasty revealed the good improvement of the total TUG time after six months post-surgery, but also showed residual differences between phases [20]. Obtaining the same time during the TUG test of patients in different diseases only indicates how far their results differ from the results of healthy children. Patients with the same times may have different percentages in the various components of the TUG test. The total test time does not indicate those components that the patients find most difficult to deal with and what their functional deficit consists of. In this work, for the first time, the percentage analysis of the components of the TUG test was applied to differentiate the functional abilities of the studied groups of patients and to find important causes of their motor problems. The analysis of the percentage of the individual components of the TUG test demonstrated significant differences between the studied subgroups of patients and thus it renders it possible to program individual physiotherapy focused on current issues.

In the group of healthy children, the total duration of the TUG test was 5.08 s. The lowest percentage share was revealed in the following components: tug1% and tug5%, while the highest was in the tug4% component. In the examined diseases, as compared to the control group, lower values were observed for the following components: tug1% and tug2%, while higher values of the percentages of tug3%, tug4% and tug5% in the TUG test, and that in all groups of patients. Among the analyzed disease groups, the total time of performing the TUG test was closest to the control group in patients with scoliosis (5.24 s). The longest time to perform the TUG test was measured in the group of patients with NP1 (6.88 s).

Patients with scoliosis had the greatest difficulty in completing the final section tug4% (29.64%), the initial section tug2% (22.36%) and the turn tug3% (20.21%). The reasons for slow walking can be seen in the three-dimensional deformation of their spine or incorrect positioning of the pelvis. Although patients with scoliosis do not have problems with walking, clinical observations suggest some stiffness and asymmetry of their gait [21]. Wang et al. [22] proved the existence of a relationship between the angle of primary curvature and the amount of pelvic rotation in the transverse plane and their impact on gait. In the studies by Gao et al. [23] people with scoliosis recorded a significantly longer time to perform the TUG test compared to healthy individuals (6.0 s), while patients with severe scoliosis performed the test faster (6.5 s) than patients with moderate (6.9 s) and light (6.8 s) spinal scoliosis. Rapp et al. [24] suggest that the size of pelvic asymmetry and the angle of curvature may change the kinematic features of gait, and hence induce problems with turning and walking straight sections. In turn, Syczewska et al. [25] found that the amount of curvature may not cause significant differences in gait due to the produced compensation mechanisms appearing in body posture.

In the group of morbidly obese patients, the highest percentage in the TUG test was occupied by turn tug3% (20.58%) and sitting down tug5% (17.24%). Incorrect body proportions and the distribution of adipose tissue reduce the mobility and fitness of patients with morbid obesity [26]. These findings are confirmed in previous reports by Ling et al. [27], who suggests that the body’s dimensions have a significant influence on the time of the TUG test (10.32 s). This would mean that the increased hip circumference, protruding positioning of the upper limbs forced the patients to move further away from the obstacle, slow down the pace of the walk and cover a longer distance at the turn, and then brake again before turning and sitting down.

In severely obese patients, muscle fibers (type IIb) associated with short-term, high-intensity, low-repetition activities predominate, which allows them to generate high levels of strength for a short period of time [28]. Due to the specific structure of their body, the increase in swaying sideways during walking, paradoxically, could support the speed of straight walk in this group of patients [29]. 

Similarly, to the group of obese children, in the group of NP1 patients, we found the highest percentages of TUG test components during the turn—tug3% (20.21%) and sitting down—tug5% (16.79%). These children also recorded the longest duration of the complete TUG test (6.88 s). The reason could be problems with balance in high positions while walking [30], reduced muscle strength or coordination when performing complex movements [31]. In our own research, in the NP1 group, patients with CP had the longest TUG test time of 7.42 s. In children with CP, gait abnormalities were a consequence of neurological dysfunctions of the musculoskeletal system in the form of: asymmetric joint contractures, muscle spasticity and abnormal phasic muscle activity [32]. Abnormalities in one of the major joints in the lower limb usually had an impact on the function of the other joints in the support limb. The test times obtained were similar to the results of other studies of CP patients with GMFCS I/II (7.6 s) [33]. At the analyzed stage of treatment, the large dispersion of the test results of the studied groups could indicate significantly different functional abilities of patients, resulting from the course of the disease, as well as the individual speed of recovery of lost functions in the process of physiotherapy.

Comparing the individual percentages of the components of the TUG test in the NP2 group, the largest share was found during getting up—tug1% (11.35%), the initial section—tug2% (24.61%) and the final one—tug4% (29.85). In the observations of Hoffman [34], patients after brain tumor surgery performed the TUG test within 5.1 s, and their recorded test time was very similar to the time obtained in the group of healthy children. On the other hand, in the group of children after stroke, Berkner et al. [35] observed a slower gait, shorter stride and increased walking cadence which results from difficulties with tasks requiring greater coordination [36,37]. A study by Rahman et al. [38] in patients after TBI with the TUG test demonstrated a significantly reduced walking speed (11.5 s) and frequency of steps, when compared to healthy children. In the studied group of NP2 patients, there was a large dispersion of the TUG test results, which resulted from the different clinical conditions of the patients. Neuroplastic changes in children after brain injury are different than in adults, because children still experience developmental maturation of the brain, at the same time experiencing neuronal changes related to trauma or physiotherapy [39]. Systematic, and early onset physiotherapy allows for the return of lost motor functions in most patients with neurological dysfunction [40,41]. Similar results were observed in the group of patients with OI. Comparing the individual percentages of the components of the TUG test in relation to the studied groups in patients with type I OI, the largest share concerned: the final section—tug4% (30.49%), the initial section—tug1% (24.34%) and sitting down—tug5% (18.08%) [42]. The past fractures and reduced muscle strength of the lower limbs meant that the patients’ gait was slower than in healthy children. This is confirmed by reports on gait, which suggest that type I OI patients walk similarly to their healthy peers, but the quality of their gait differs significantly from that of healthy children. The gait of children suffering from OI is characterized by an increased time of double support, a delay in detachment of heel from the ground, a decrease in the range of motion in the ankle joint, including the amount of plantar flexion of the foot and the force of plantar flexion during pushing away compared to their healthy peers of comparable age [43]. Despite their limited physical activity, patients with type I OI are much stronger and more functional than those with other types of OI [44]. In the Daniels and Worthingham test, the mean strength of the plantar flexors for the OI group (3.9 ± 1.2) was lower than the average for the control group (5.0 ± 0), which may translate into their lower gait dynamics. Similar results were obtained with the use of the Biodex System 3 system, where healthy children recorded greater strength of plantar flexion in both the left and right ankle joints compared to patients from the OI group [45]. In the case of OI patients, the disease has a very individual course, fear of falling, reduced efficiency, weakened muscle strength or the level of skeletal mineralization in the studied OI population may be the cause of slower walking and decreased cadence of walk [46]. The discrepancies between the studies in the TUG test could be partly due to significant differences between the groups of patients in terms of their age, body mass, or height. The final TUG result may have been influenced by the degree of damage resulting in various neurological and orthopedic consequences. The use of the percentage share of individual components of the TUG test proved that each of the studied groups of patients is characterized by different severity of functional problems. Due to the lack of previous approaches of this type to patient assessment, it is difficult to say how far the disease itself or the physiotherapy carried out changed the functional status of patients. Further research must be carried out to increase the number of patients in the respective groups, and thus obtain a more homogeneous composition of patients. The program for physiotherapy and monitoring the progress in improvement should consider the difficulties occurring in individual patients or groups of patients, which resulted from the analysis of the percentage share of the TUG test components. 

The obtained results of this study can have implications for clinical practice. The TUG test consists of several functional tasks: walk, turn, sit-to-stand and stand-to-sit. The separate measurements of these components enable identification of the patients’ functional problems. Each of the examined groups had different time participations of the TUG components. The monitoring of the changes in TUG components participation in the test together with changes of the total TUG time can be additional mean of the assessment of the efficacy of rehabilitation treatment.

The main limitation of the study is the variability of the TUG test. The coefficient of variability for the total TUG time was low: around 3%, for the TUG phases the mean coefficient of variation was higher: 7.1%. In our study the measurements were performed by one physiotherapist, but in clinical every day work the assessments were usually conducted by different staff members. This could be an additional variability factor, although in the previous study [15] the inter-rater ICC were high for the total TUG time: 0.992, and from 0.86 to 0.992 for TUG phases.

## 5. Conclusions

The assessment of the gait and balance functions based on the analysis of the percentage of individual components of the TUG test proved to be a sound diagnostic tool, differentiating the studied groups of patients. The analysis of the components of the TUG test can assist therapists and doctors in the assessment of the mobility of children and adolescents, monitoring the effects of physiotherapy or the effects of surgical procedures.

## Figures and Tables

**Table 1 ijerph-19-04602-t001:** Clinical characteristics of the examined group.

Groups	Age [years] Med. Min–Max	Body Mass [kg] Med. Min–Max	Body Height [cm] Med. Min–Max	Paresis sin/dx	4-Limb Paresis or Paresis of Lowe Limbs	Musculoskeletal Deformities
N	12.4	45	155.5	N/A	N/A	N/A
	6.9–18	19–85	111–193			
Sc	14.1	47.9	160	N/A	N/A	The spine sc.11–40 deg., pelvic deform, asym. of l. limbs, def. of feet
	6.7–17.7	20–103	120–187			
BT	15.1	59	161			Postural defect, segment asymmetries pelvic def., asym. of l. limbs, def. of feet
	7.4–17.6	28–117	135–181	10/8	8/1	
CVA	11.9	47	160			Postural defect, segment asymmetries, pelvic def., asym. of l. limbs, def. of feet
	8.1–16.2	22–75	125–174	2/3	1/8	
TBI	16.6	64	165			Postural defect, segment asymmetries, pelvic def., asym. of l. limbs, def. of feet
	9.8–17.2	32–99	142–181	2/6	4/3	
GBS	12.1	45	157	N/A		Postural defect, segment asymmetries, pelvic def., asym. of l. limbs, def. of feet
	6.8–18	25–90	127–178		13/3	
CP	14.4	52	160			Spine sc.10–30 deg., pelvic def., asym. of l. limb axes, def. of feet, contractures
	7.6–17.9	22–81	124–188	9/7	3/5	
OB	16.1	124	169	N/A	N/A	Postural defects, sc.10–20 deg., Disturbances of the axis of l. limbs, def. of feet
	9.1–18	88.7–173	148–187			
OI	14.5	39.2	150.7	N/A	N/A	Spine: sc.10–50 deg. pelvic def., chest def., disturbed axis of u. and l. limbs, def. of feet
	5–18	12.5–78	85–177			

N/A—not applicable, N—control group, Sc—scoliosis, BT—brain tumor, CVA—post—stroke, TBI—traumatic brain injury, GBS—Guillain–Barré syndrome, CP—cerebral palsy, OB—morbid obesity and OI—osteogenesis imperfecta type I.

**Table 2 ijerph-19-04602-t002:** The results of the individual components and the total time of the TUG test in reference and in the studied groups of patients.

Groups	Times of the TUG Test and Individual Test Components					
	Tug1 (s)	Tug2 (s)	Tug3 (s)	Tug4 (s)	Tug5 (s)	TUG (s)
N	0.73	1.35	0.81	1.44	0.71	5.08
*n* = 244	0.43–1.04	0.55–2.35	0.4–1.54	0.6–2.27	0.31–1.48	3.75–6.98
Sc	0.59	1.17	1.0	1.54	0.84	5.24
*n* = 164	0.46–0.98	0.84–1.98	0.81–1.68	1.14–2.16	0.48–1.79	3.92–7.31
NP1	0.78	1.75	1.28	2.03	1.06	6.88
*n* = 43	0.52–2.28	1.01–5.68	0.98–3.09	1.46–6.24	0.64–3.52	4.93–19.87
NP2	0.68	1.42	1.24	1.81	1.1	6.06
*n* = 59	0.55–1.6	0.86–3.99	0.97–2.46	1.4–4.63	0.55–2.76	4.56–13.89
OB	0.64	1.29	1.21	1.75	0.97	5.95
*n* = 72	0.53–1.21	0.96–1.8	1.0–1.59	1.37–2.09	0.68–1.68	4.73–7.66
OI	0.63	1.49	1.11	1.74	1.03	6.13
*n* = 34	0.46–1.62	1.17–2.08	0.62–1.91	1.14–2.47	0.66–1.77	4.95–8.75

**Table 3 ijerph-19-04602-t003:** Percentage share in the individual components of the TUG test among the studied groups.

Components of TUG Test	Groups	Percentage of the Individual Components of the TUG Test						
		Median	Min	Max	Lower Quartile	Upper Quartile	10th Percentile	90th Percentile
Tug1%	N	14.28	8.68	22.81	12.42	16.80	10.96	18.32
Tug2%		27.02	11.65	44.33	24.42	29.66	22.09	31.49
Tug3%		16.71	7.87	32.69	14.49	18.29	12.97	19.86
Tug4%		28.75	17.54	37.12	26.94	30.73	24.45	32.64
Tug5%		14.28	7.42	27.06	12.09	16.14	10.68	18.15
Tug1%	Sc	11.28	7.46	16.79	10.53	12.35	9.97	13.07
Tug2%		22.36	17.53	28.93	21.26	23.82	20.25	25.13
Tug3%		20.21	14.50	30.32	19.35	21.49	18.15	22.57
Tug4%		29.64	24.09	33.89	28.47	30.93	27.16	31.89
Tug5%		15.68	10.11	27.71	14.23	17.24	13.23	18.64
Tug1%	NP1	11.13	7.82	13.48	10.11	11.80	9.09	12.31
Tug2%		22.41	17.53	29.53	21.15	24.51	19.93	25.71
Tug3%		20.21	15.43	26.97	18.32	21.52	17.22	22.71
Tug4%		28.86	24.47	34.27	27.76	30.98	26.98	32.35
Tug5%		16.79	9.55	27.65	13.98	19.13	12.45	21.41
Tug1%	NP2	11.37	7.63	14.99	10.19	12.70	9.21	13.12
Tug2%		24.61	18.28	32.49	22.33	25.98	20.89	27.22
Tug3%		18.83	10.06	22.64	17.17	20.49	16.12	21.22
Tug4%		29.85	25.51	44.97	28.02	31.19	26.94	33.27
Tug5%		15.58	9.74	26.58	13.32	17.22	10.98	20.15
Tug1%	OB	10.87	8.61	18.14	10.20	11.59	9.40	12.65
Tug2%		21.89	18.91	28.35	20.58	22.89	20.07	24.45
Tug3%		20.58	14.88	23.13	19.53	21.72	18.40	23.14
Tug4%		29.29	23.22	34.22	27.70	30.42	25.95	31.68
Tug5%		17.24	12.27	23.49	15.38	18.65	14.18	20.86
Tug1%	OI	10.22	7.80	22.03	9.51	13.09	8.72	19.26
Tug2%		24.34	19.03	30.71	21.93	26,91	19.77	29.37
Tug3%		18.13	11.33	21.84	14.49	19.05	11.83	20.79
Tug4%		30.49	19.37	37.36	28.69	31.84	19.98	33.39
Tug5%		18.08	12.22	21.87	15.61	19.01	14.16	20.52

**Table 4 ijerph-19-04602-t004:** The significance of differences in the individual components of the TUG test between the studied groups.

Groups	Sc	NP1	NP2	OB	OI
	Getting up Tug1%				
N	0.001	0.001	0.001	0.001	0.001
	Initial pass Tug2%				
N	0.001	0.001	0.01	0.001	0.001
Sc			0.002		
OB			0.001		0.03
	Turn Tug3%				
N	0.001	0.001	0.001	0.001	
Sc			0.004		0.001
NP1					0.001
OB			0.002		0.001
	Final pass Tug4%				
N	0.01				
	Sitting Tug5%				
N	0.001	0.001		0.001	0.001
Sc				0.01	
NP2				0.02	

## Data Availability

The datasets during and/or analysed during the current study are available from the corresponding author on reasonable request.

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
