# Peer review of "Using the TUG Test for the Functional Assessment of Patients with Selected Disorders"

_ijerph, 2022, doi:10.3390/ijerph19084602_

Round 1

Reviewer 1 Report

 Thank you for a very interesting study.

The authors stated that "The research was carried out by the same thera- 86
pist measuring the time of each component of the test (tug1-tug5) on a Samsung Galaxy 87A70 mobile phone with an accuracy of hundredths of a second".

-Is this a valid and reliable measure? If so, please add a reference. 

Author Response

We would like to thank the Reviewers for such insightful comments regarding both the overall presentation of our study in the paper and on many details regarding how the information is presented. We therefore present a thoroughly revamped draft (all the changes are marked in the paper by the bolded text, and we also answered all the remarks on the point-to-point basis) in which we have sought to clear up all inconsistencies and to address all the issues raised by the reviewers.

  1. The authors stated that "The research was carried out by the same therapist measuring the time of each component of the test (tug1-tug5) on a Samsung Galaxy 87A70 mobile phone with an accuracy of hundredths of a second".

-Is this a valid and reliable measure? If so, please add a reference. 

The following sentence was added to the Methods section: “The use of smartphone applications in time measurements in clinical settings is gaining popularity, as they demonstrate high accuracy and consistence with the traditional, lab-based methods [Chan et al., 2016, Lein et al., 2019].

Reviewer 2 Report

This manuscript studies the percentage of the individual components of the Timed Up-and-Go (TUG) test in functional diagnostics in the population of healthy children and in clinical trials of patients with various diseases.

The empirical findings provide some implications helpful for practical aspects and confirm TUG test as a good diagnostic tool. Conclusions about the present investigation are reported.

I have reviewed this manuscript, but I am sorry to say that the study present severe limitations, I have doubts about the technical quality of this work. Therefore, with regret, I must recommend its rejection.

In order to improve the presentation of their manuscript, the authors maybe wish to consider the following few comments:

  • The test was performed five times for each patient and the averages of each component were used in the analysis. In this way the authors completely ignore the variability of the measures, which must be considered in the study.
  • The percentages of the test component are analyzed individually. In this way the authors completely ignore their joint multivariate distribution, namely (tug1, tug2, …,tug5) form a composition subject to a constrain, i.e. the total time. A methodology coherent with the nature of the data in anlysis should be considered.   A possible reference could be Dumuid D, Stanford TE, et.al. Compositional data analysis for physical activity, sedentary time and sleep research. Stat Methods Med Res. 2018 Dec;27(12):3726-3738.
  • I have wondered the data cleaning processes of authors. They should write their data cleaning processes, if any, in detail in the methodology section.

Author Response

We would like to thank the Reviewers for such insightful comments regarding both the overall presentation of our study in the paper and on many details regarding how the information is presented. We therefore present a thoroughly revamped draft (all the changes are marked in the paper by the bolded text, and we also answered all the remarks on the point-to-point basis) in which we have sought to clear up all inconsistencies and to address all the issues raised by the reviewers.

  1. The test was performed five times for each patient and the averages of each component were used in the analysis. In this way the authors completely ignore the variability of the measures, which must be considered in the study.

The individual variability of the patients’ results depends on the several factors, both from patient’s side, and the examiner’s sides. The repetitions and averaging were used to eliminate accidental changes. For the purpose of this review two patients, one with scoliosis and one with hemiparesis. The measurements were taken within three days, seven repetitions all together, performed by one physiotherapist, the same who did the original measurements. The coefficient of variability was very low for total TUG time: 2.7 % and 3.26 %, for the five phases its range was from 3.85 % to 14.95 %. The mean coefficient of variability (for TUG phases) was 7.1 %. The following sentence was added to the Discussion section: “The main limitation of the study is the variability of the TUG test. The coefficient of variability for the total TUG time was low: around 3 %, for the TUG phases the mean coefficient of variation was higher: 7.1 %. In our study the measurements were performed by one physiotherapist, but in clinical every day work the assessments were usually done by different staff members. This could be an additional variability factor, although in the previous study [Graff et al., 2018] the inter-rater ICC were high for the total TUG time: 0.992, and from 0.86 to 0.992 for TUG phases.

  1. The percentages of the test component are analyzed individually. In this way the authors completely ignore their joint multivariate distribution, namely (tug1, tug2, …,tug5) form a composition subject to a constrain, i.e. the total time.

The TUG test was designed and used as a whole, but the introduction of the new technical possibilities (IMUs, or mobile apps) enabled the analysis of the TUG phases. There are studies in which the five phases of TUG test (sit-to-stand, walk1, turn, walk 2, stand-to-sit) are measured and analysed together with the total time of the test. But up to now these studies were done on adult patients populations. They proved that not only the total TUG test time, but also the duration of the phases could be meaningful and important. The following sentences were added to the Discussion section: “The study of Caronni et al. [Caronni et al. 2018] revealed, that in adult neurological patients the turning phase of the TUG test highly correlated with the balance problems, and was a good predictor of the Mini-BESTest scale result. In the obese adult women the walk and turning phases as well as the total TUG time are increased in respect to the women with normal weight [Cimolin et al. 2019]. The functional assessment of patients after hip arthroplasty revealed the good improvement of the total TUG time after six months post-surgery, but also showed residual differences between phases [Gasparutto et al. 2021].”

  1. A methodology coherent with the nature of the data in anlysis should be considered.   A possible reference could be Dumuid D, Stanford TE, et.al. Compositional data analysis for physical activity, sedentary time and sleep research. Stat Methods Med Res. 2018 Dec;27(12):3726-3738.

We would like to thank the reviewer for the indication of this interesting paper, which was not known to us before. We for sure will use the described in it method for future studies, but in our opinion it will be difficult to use it for our study. This paper describes the statistical approach for looking of the dependencies between time series data and other factors (in the example in this paper – children obesity). Our study shows differences of the components participation in the total TUG time in five patients’ groups and in healthy controls.

  1. I have wondered the data cleaning processes of authors. They should write their data cleaning processes, if any, in detail in the methodology section.

The data were not cleaned, the raw data were taken to the analysis. The patients were instructed before the measurements about the purpose of the test, and how to perform it. In some cases the patient had some hesitation during first performance, and these data were discarded, and subsequent repetitions were done.

Round 2

Reviewer 2 Report

I would like to thank the authors for their reply.

I have reviewed this manuscript, but I am sorry to say that the study presents severe limitations that the authors did not fill in their revision.

In my opinion the two paragraphs added to highlight the limitations of the study are not sufficient to sensibly improve the technical quality of this work. A complete revision of the analysis methods should be implemented in order to honor the multivariate distribution of the data and their correlation structure.

Therefore, with regret, I must recommend its rejection.

Author Response

We wish to thank the reviewers and editors for the time spent considering the draft in this revised form.

However, we are very disappointed with Reviewer 2’s conclusion to recommend that the paper be rejected for publication on the stated grounds, and would like to take the opportunity to explain on what grounds we entirely disagree with this reviewer’s conclusion that a “complete revision of the analysis methods should be implemented in order to honour the multivariate distribution of the data and their correlation structure”. As such, we trust that we will persuade the editors and Reviewer 2 to reconsider the rejection recommendation.

Firstly, please note that main aim of this study is to see if the phases of the TUG test (sit-to stand, walk 1, turn, walk 2, stand-to-sit) differ among selected diseases and healthy subjects in paediatric population. Similar studies have been performed in some of the adult population, but to our knowledge not among children and adolescents. Let us be explicit that we were not seeking to identify or report any correlations or other dependencies between the TUG time series any other variables. Rather, our study is a descriptive one. Although it may seem simple, the study, in our view, does have a very serious clinical implication: it allows for better understanding of the patients’ disability and gives the clinical staff a relatively simple tool to evaluate and monitor changes in functional status of the patients. Osteogenesis imperfecta and paediatric Guillain-Barre syndrome are rare diseases, but due to the fact that the data were collected at Poland’s largest paediatric hospital, we were able to collect sufficient data – which may therefore prove beneficial for other centres, where such patients are treated sporadically.

Our viewpoint is that statistics is ultimately always just a tool, and its use has to be carefully tailored according to the aim and purpose of any particular study. We cannot see any justification for the use of multivariate methods in our study. These methods are used in many disciplines, but in all of them the aim of their use is to make a model for better reproduction of certain changes and to use such a model in making future predictions [for example Cao et al., https://doi.org/10.1016/S0167-2789(98)00151-1, Batal et al. https://doi.org/10.1145/2339530.2339578]. In this particular study, how might we construct a model for a TUG time series dependent on different diseases?

Perhaps it is also worth pointing out that in some of our own previous studies we have used quite advanced statistical and modelling methods [e.g. Syczewska et al. https://doi.org/10.1016/j.bbe.2020.07.002, Syczewska et al. https://doi.org/10.1016/j.bspc.2021.102496], and so it is not the case that we are unfamiliar with how to apply the kind of multivariate methods Reviewer 2 has insisting upon. In each of our previous publications, however, statistical and modelling methods were treated as tools appropriately selected for answering the research questions specifically being addressed. We are convinced that the tools employed in the present paper are entirely adequate for the stated objective.

In conclusion, we would encourage both the editors of the journal and Reviewer 2 in specific to reconsider the latter’s rejection recommendation, and instead reconsider accepting the paper for publication. 
